# HIPI: Spatially resolved multiplexed protein expression inferred from H&E WSIs

**Ron Zeira** *, **Leon Anavy, Zohar Yakhini, Ehud Rivlin, Daniel Freedman**

Verily AI, Tel Aviv, Israel

* ronzeira@gmail.com

## Abstract

Solid tumors are characterized by complex interactions between the tumor, the immune system and the microenvironment. These interactions and intra-tumor variations have both diagnostic and prognostic significance and implications. However, quantifying the underlying processes in patient samples requires expensive and complicated molecular experiments. In contrast, H&E staining is typically performed as part of the routine standard process, and is very cheap. Here we present HIPI (H&E Image Interpretation and Protein Expression Inference) for predicting cell marker expression from tumor H&E images. We process paired H&E and CyCIF images taken from serial sections of colorectal cancers to train our model. We show that our model accurately predicts the spatial distribution of several important cell markers, on both held-out tumor regions as well as new tumor samples taken from different patients. Moreover, using only the tissue image morphology, HIPI is able to colocalize the interactions between different cell types, further demonstrating its potential clinical significance.

**Data Availability Statement:** Raw H&E images, raw CyCIF images and processed CyCIF cell data can be obtained from github.com/labsyspharm/CRC_atlas_2022. Code for processing the data,

## Author summary

We present HIPI (H&E Image Interpretation and Protein Expression Inference) for predicting cell marker expression from tumor H&E images. We process paired H&E and CyCIF images taken from serial sections of colorectal cancers to train our model. We show that our model accurately predicts the spatial distribution of several important cell markers, on both held-out tumor regions as well as new tumor samples taken from different patients. Moreover, using only the tissue image morphology, HIPI is able to colocalize the interactions between different cell types, further demonstrating its potential clinical significance.

## 1 Introduction

Analysis of histopathological images from stained tissue sections has played a key role in identifying tumor features with diagnostic and prognostic significance [1]. Stained tumor inspection enables the characterization of intra-tumor regions and of tumor interactions with the microenvironment and the immune system. Moreover, it facilitates the detection of predictive

training and evaluating the model is in https://github.com/RonZeira/HIPI/.

**Funding:** The author(s) received no specific funding for this work.

**Competing interests:** I have read the journal's policy and the authors of this manuscript have the following competing interests: All authors are employed by Verily Life Sciences.

biomarkers for disease progression and treatment [2]. Hematoxylin and eosin (*H&E*) staining of tumor tissues is routinely performed as part of cancer standard care [3]. Using H&E stained Whole Slide Images (*WSIs*) of a tumor, pathologists can distinguish between the nuclear and cytoplasmic parts of cells, and identify patterns in tissue structure and in the cell distribution. H&E staining is often complemented by immunohistochemistry (*IHC*) staining of additional tissue slices which allows the identification and quantification of cancer and immune-specific biomarkers [4]. However, IHC staining is a more delicate, time consuming and expensive process; and therefore, the number of IHC stains performed on each sample is limited. Furthermore, not all cellular molecular biomarkers can be detected using IHC staining.

Spatial molecular methods have been emerging in recent years as a powerful tool for measuring cellular biomarkers while retaining spatial information and highlighting spatial relationships [5]. Such approaches quantify the expression of different molecular modalities together with their spatial locations, providing a rich molecular image of the tissue that may not always be obtained with traditional staining methods. Data acquired using spatial technologies varies in terms of the type of modality (RNAs, proteins, metabolites), spatial resolution (sub-cellular, cellular, a few cells) and the number of measured features (a few targeted features, genome/transcriptome-wide measurements) [6]. For instance, *Spatial Transcriptomics* measures the RNA expression of all genes across thousands of spots on a tissue slice, each containing around a dozen cells [7]. In a different approach, the cyclic immunofluorescence (*CyCIF*) technology creates multiplexed images of targeted protein expression and enables the detection of cells and cellular expression [8]. Nevertheless, spatial technologies have mainly been used for exploratory cancer research, while their application for routine care is very limited due to the specialized equipment and cost required [5].

Computational pathology methods for the analysis of histopathology images have resulted in significant interest and progress recently [9, 10]. Artificial intelligence models have been trained to facilitate various tasks that previously required manual labor and expert annotation, as well as novel tasks that were not manually possible. For example, models have been developed to predict prognostic and diagnostic labels from sample WSIs such as cancer staging, grading, classification, sub-typing, survival prediction, treatment response, biomarker detection, etc. [11, 12]. Another class of models, referred to as *virtual staining*, are trained to transform one type of stained tissue WSI into a different type of stain of the same tissue, reducing the need to actually stain multiple tissues [13, 14]. More recent models allow for the transformation of histopathology images into spatial molecular maps, such as RNA expression measured using spatial transcriptomics [15–19]. However, training these models is difficult, as datasets of paired images and spatial molecular data from the same tissue are scarce. Therefore, these models were evaluated only on slices coming from the same tissue used for training but were mostly unsuccessful generalizing to new tissue samples [15–17]. A very recent article proposed a virtual staining model to map H&E into virtual CyCIF stains [20], but only evaluated on slides coming from the same tissue.

Here we propose an artificial intelligence model called **H**&E Image **I**nterpretation and **P**rotein Expression **I**nference (**HIPI**) to predict multiplexed CyCIF protein expression levels from whole slide H&E images. We align H&E and CyCIF data taken from adjacent tissue slices of colorectal cancer in order to train our model and predict multiple cellular markers. To cope with data scarcity, we base our model on recent advancements in Self-Supervised Learning (*SSL*) that train on large amounts of unlabeled data to extract meaningful image representations [21–23]. We further use an augmentation scheme during training to address staining intensities and other artifacts specific for H&E images.

We show that HIPI accurately predicts protein expression levels on both held-out colorectal cancer tissue regions and on new tumor samples taken from different patients not used during

training. Our method is able to detect the occurrence and co-occurrence of important molecular markers based only on image morphology from routinely taken H&E WSIs. For instance, the interaction between PD1 and PDL1 markers predicted by our model can be therapeutically targeted in colorectal cancer patients and thus is clinically significant [24]. Moreover, our model enables the generation, or inference, of spatial maps for multiple markers. This inference of vector valued markers would be too expensive to generate with traditional IHC stains.

## 2 Materials and methods

### 2.1 Predicting CyCIF protein expression levels from H&E images

We developed **H**&E Image **I**nterpretation and **P**rotein Expression **I**nference (**HIPI**), a deep learning model for predicting CyCIF protein expression levels from whole slide H&E images (Fig 1A and S1 Appendix). HIPI predicts the expression levels of multiple proteins from a single H&E image, operating on image tiles. Each tile is fed into a feature extraction Vision Transformer model pretrained on many histological images with self supervision [23]. The resulting feature vector of size $d = 384$ per tile is used as the input for a four-layer fully connected regression head that outputs the expression values of various proteins. The protein expression of each tile is then reordered (de-tiled) to give tissue-wide protein expression maps.

We trained HIPI on a subset of tiles from 22 pairs of CyCIF and H&E Whole Slide Images (WSIs) taken from adjacent serial slices of an adenocarcinoma patient obtained from the

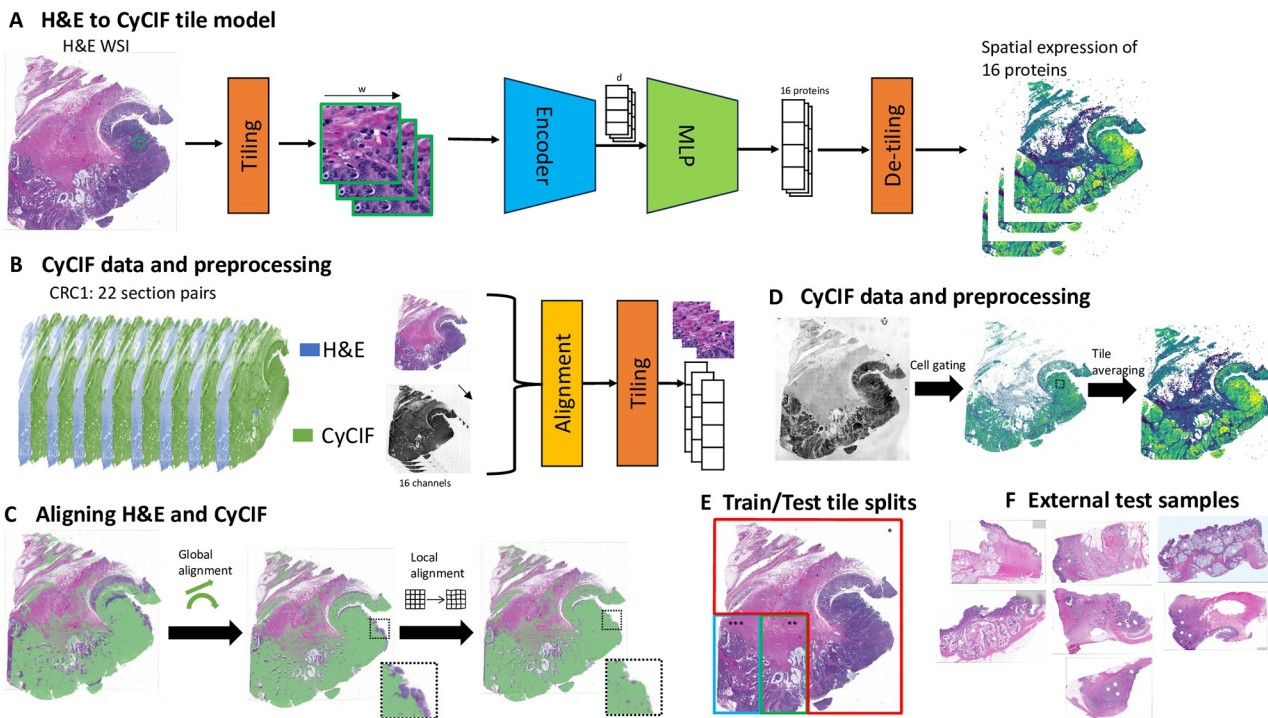

**Fig 1. H**&E Image **I**nterpretation and **P**rotein Expression **I**nference—overview of the data and method: (A) A schematic overiew of HIPI. An H&E slide is processed in tiles through a Deep-Learning prediction model generating multiplexed spatial expression level of 16 proteins. (B) An overview of the data and preprocessing steps. Pairs of H&E and CyCIF images taken from adjacent tissue slices are aligned and processed to generate image tiles with corresponding expression levels. (C) Alignment of adjacent H&E and CyCIF images using global linear transformation followed by local non-linear registration. The inset is a zoom-in on a tile to demonstrate the effect of the local refined alignment. (D) Deriving tile level expression from CyCIF cell data. Cell locations and expression were taken from [25]. The expression is then aggregated at the tile level. The figure illustrates expression of Ki67 marker on slice 25 (yellow—high expression). (E) Train (* red), validation (** green) and test (*** blue) data splits for sample CRC1 illustrated on a single slice. (F) Seven additional CRC samples from different patients.

literature (Fig 1B) [25]. This is the sample referred to as CRC01 in [25] as well as in this work. We used 16 measurements that were used by [25] as cellular markers: cytokeratin, Ki-67, CD3, CD20, CD45RO, CD4 CD8a, CD68 CD163, FOXP3, PD1, PDL1, CD31, $\alpha$-SMA, desmin, and CD45. The training of HIPI required matching images of H&E with images of the aforementioned CyCIF protein levels. Direct mapping of cells between adjacent tissue slices is challenging due to both biological and technical variation. Therefore, we used a two-step registration process of adjacent CyCIF and H&E image pairs. First we performed a global alignment of the entire CyCIF image onto the H&E image using an affine transformation. Second, the alignment was fine-tuned using non-linear tile level local alignment (Fig 1C and S1 Appendix). The model was then trained on tile-level protein expression for tiles of size $256 \times 256$ pixels, each corresponding to roughly a $128\mu m \times 128\mu m$ tissue tile. The tile level protein expression is calculated by aggregating all corresponding cell calls from the CyCIF data (Fig 1D and S1 Appendix).

We trained the model using 75% of the tiles coming from the same three quadrants of the 22 tissue WSIs while the tiles in the bottom left quadrant of each slide were left out as test data (Fig 1E). Out of the bottom left quadrant tiles, the right half was used for validation and assessment of the model during training while the left half was left out for assessment of the final model as test data. With this partition, we make sure that tiles used for training and testing come from different areas of the tissue, avoiding memorization and eliminating spatial effects on the results. This is in contrast to previous models that either evaluated on adjacent slices from the same tissue or randomly split regions of the same tissue into training and test sets [15–17, 20]. Overall, the training set had 1,351,680 tiles, the validation set had 388,783 tiles and the test set had 379,142 tiles. During training we used an image augmentation scheme designed specifically for pathology images to account for artifacts such as image quality and staining intensity and enable better generalization [26].

## 3 Results

### 3.1 HIPI predicts CyCIF from H&E

We trained HIPI to predict CyCIF protein expression from H&E image tiles from CRC01 and used the model to predict protein markers on all image tiles (Section 2.1). Overall, we observe that our model's predictions are highly correlated to the measured expression with median Pearson correlation of 0.66, 0.61, 0.64 for the training, validation and test tiles, respectively, across all slice pairs and proteins (Fig 2 and S1 Fig). Moreover, on most markers we do not see significant differences in prediction performance between tiles used for training and the tiles left out for testing. Between the different slice pairs, we see similar levels of prediction correlation for a given protein. For some markers, e.g. Keratin, Ki67 or CD45, the model shows high concordance with the measurements with median correlations of 0.86, 0.73 and 0.67 respectively between the tiles of the different slice pairs (Fig 2). Furthermore, the model is especially successful at capturing regions with high marker expression. To show this, we calculated the *top-X% accuracy* defined as the overlap percentage between the highest X% of tiles measured for a certain marker and the highest X% of tiles predicted for the same marker. For instance, the top-20% accuracy obtained by the model in slice 96 for Keratin, Ki67 and CD45 is 72%, 60% and 65% respectively (Fig 4 and S1 Table).

On the other hand, some markers, such as PD1 or PDL1, are more difficult to correctly infer from the H&E image although the model's predictions are still positively correlated to the measured expression. The median correlation for PD1 is 0.4 with no difference between train and test sets, while for PDL1 we see a slight degradation in correlation between train and test sets. The measured values of these markers have grid like artifacts as a result of the acquisition

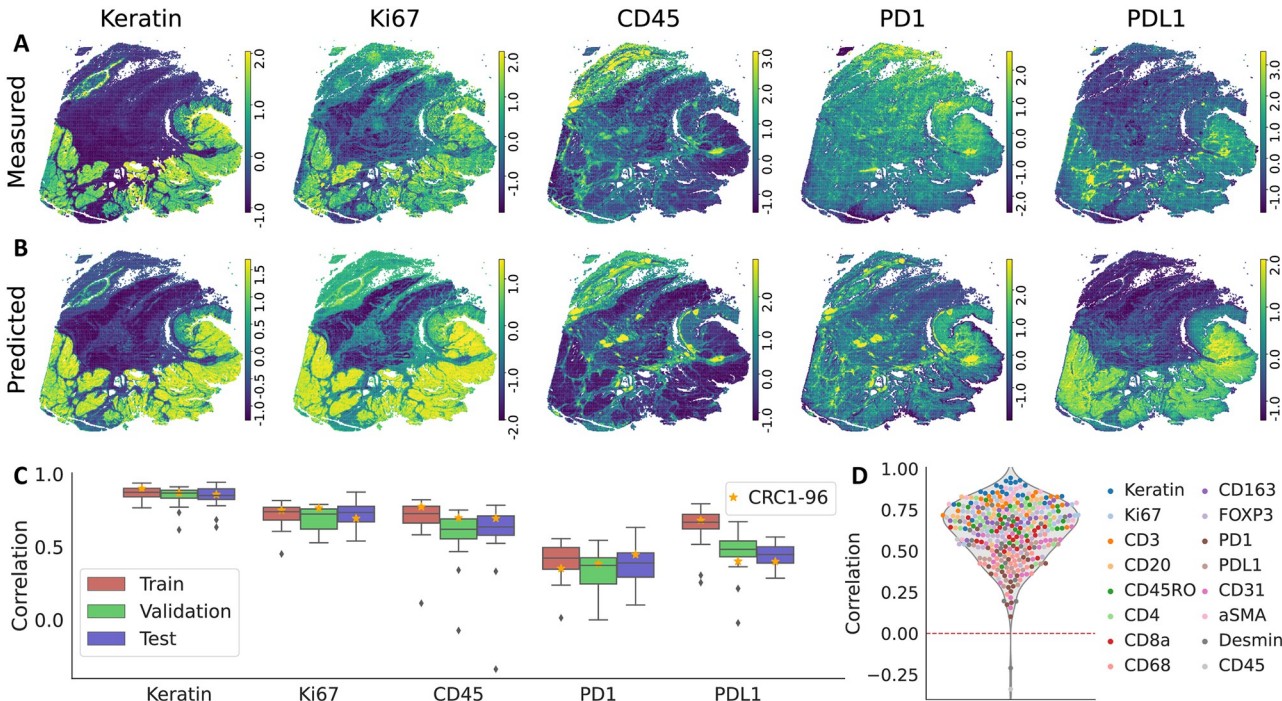

**Fig 2. Selected results on sample CRC01 slice 96 for five markers: Keratin, Ki67, CD45, PD1, PDL1.** (A) tile level CyCIF expression (z-score). (B) Prediction of our model for each tile (z-score). (C) Pearson correlation between measured and predicted values of the selected markers. We calculate the correlations for the train/validations/test tiles of each slice of sample CRC01 separately (slice 96 is marked with a star). (D) Pearson correlation between measured and predicted values across all slices and all markers on the test set.

cycles and image stitching of the CyCIF protocol [8]. This phenomenon contributes to the lower correlation between the measured and predicted values. However, we still see that the model has good performance capturing highly expressed tiles (S4 Fig and S1 Table). The top-20% accuracy for PD1 and PDL1 in slice 96 was 38% and 50% respectively.

## 3.2 HIPI generalizes to new colorectal samples

Next, we evaluated HIPI on tiles from 7 additional CRC samples that were not used for training and originated from different patients spanning different histologic and molecular subtypes (named *CRC* 2, 3, 12, 13, 14, 15 and 17 in [25], Fig 1F). These samples were selected based on the physical proximity of their H&E and CyCIF slices. We preprocessed the new samples similar to the training samples (Section 2.1). This out of distribution test dataset from the 7 new tumors yielded 794k tiles.

We see that HIPI predictions are mostly positively correlated with the marker measurements (Fig 3 and S2 Fig). Naturally, the predictive performance is lower in these out of distribution samples in comparison to the slides of patient CRC01 (S2 and S3 Figs). The median correlation between measured and predicted values across all the new slices and all markers is 0.4 in comparison to 0.64 in the test set of CRC01 (Fig 3D). Similar to the training slides from CRC01, we see considerable differences in predictive performance between markers. For example, the predictions of Ki67 and CD45 are highly correlated with the true CyCIF values across the new samples achieving levels similar to the test sets of CRC01. HIPI correctly predicts the top-20% tiles with highest Ki67 expression with 46–77% accuracy, similar to CRC01 samples (S4 Fig and S1 Table). Whereas for CD45 we observe 38–57% highly expressed tile

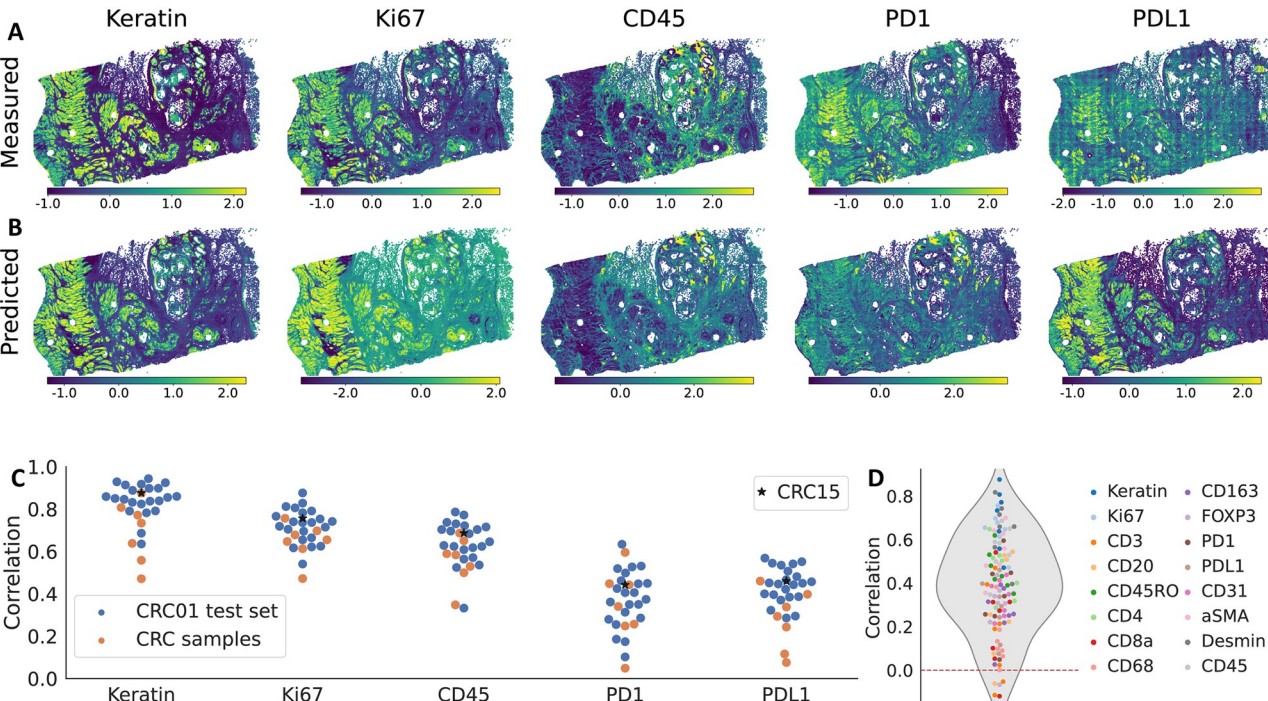

**Fig 3. Selected results on sample CRC15 for five markers: Keratin, Ki67, CD45, PD1, PDL1.** (A-B) similar to to Fig 2A and 2B. (C) Pearson correlation between measured and predicted values for the test set of each slide of sample CRC01 and all external CRC samples (sample CRC15 is marked with a star). (D) Pearson correlation between measured and predicted values across all slices and all markers on the test CRC samples.

top-20% accuracy, slightly lower than CRC01 samples. Consistent with our observations on the training samples, PD1 and PDL1 have lower levels of overall correlations. However, these levels of correlations are similar to the range of correlations observed for CRC01 test sets (S3 Fig). In addition, we see that HIPI predicts the highly expressed tiles with 29–44% and 20–50% accuracy for PD1 and PDL1 respectively (S4 Fig and S1 Table).

### 3.3 HIPI captures cell types and marker co-occurrence

Different cell types express different surface markers depending on the tissue morphology and other proximal cell types. The response of immune cells in the tumor boundary can have significantly influence disease progression and treatment response. Moreover, interactions between cells expressing certain markers such as PD1 and PDL1 can be therapeutically targeted and thus is clinically significant. It is thus important to determine the co-localization of different cell types and surface markers. Here, we analyze how tile level protein expression values capture cell surface marker expression and identify the co-localization of different cell types. Furthermore, we examine how HIPI predictions on H&E image tiles recapitulate the measured spatial relationship. We note that while our analysis is at the tile level and not in single cell resolution, it is sufficient to observe co-occurrence (e.g. PD1 and PDL1) of marker expression at the same tile to infer cell interaction.

We binarized the normalized protein expression values to obtain marker calls for each tile. To that end, we fit a two component Gaussian Mixture Model on the values of each protein and slide separately, similar to [25]. We see that the tile level marker calls capture the abundance and distribution of cell level CyCIF calls (Fig 4A in comparison to Figures 7abcdk of

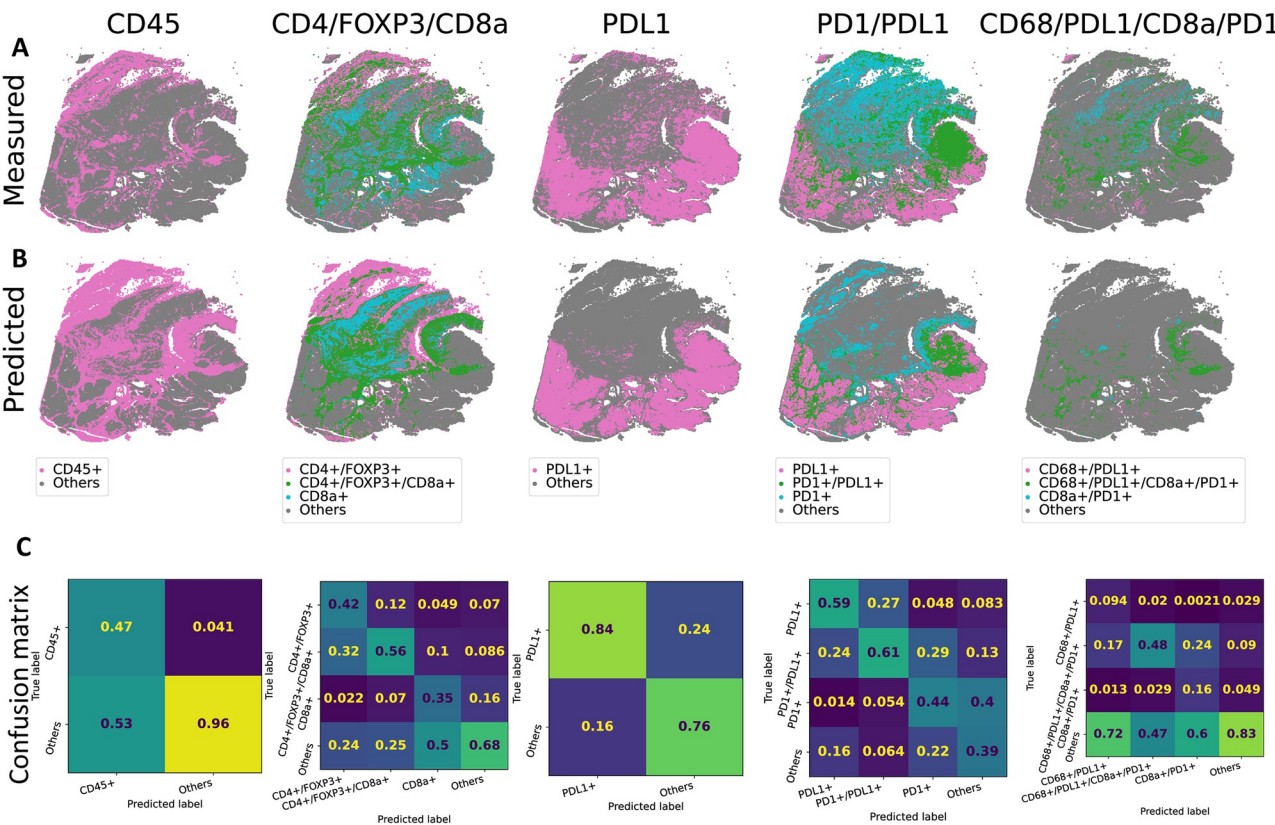

**Fig 4. Selected cell marker occurrence in CRC01–96.** (A) Occurrence and co-occurrence of CD45, CD4/FOXP3/CD8a, PDL1, PD1/PDL1 and CD68/PDL1/CD8a/PD1 in measured tile level data. (B) Predicted occurrence by HIPI. (C) Confusion matrices classifying different marker co-occurrence. Matrices are normalized column-wise by the total number of predictions for each class. Therefore, the inline numbers give the fraction of tiles predicted by the model to be in a certain class which truly belong to that class.

[25]). This shows that the tile level calls can also potentially recover the interaction or co-localization of different cell types within a tile.

Next, we evaluated the performance of HIPI in calling marker distribution and co-occurrence (Fig 4B). We see that HIPI predictions provide smoother marker calls in comparison to the measured tile level calls. For instance, we do not see the grid-like artifacts observed in the measurements of some markers such as PDL1. The model tends to be conservative and give high precision in calling the true marker occurrences, where the precision is defined as the number of true positive calls divided by the total number of predicted positive calls. That is, tiles called by the model to be marker positive are likely to be true marker positive (Fig 4C). In particular, we see that marker co-occurrences are called with high precision. For example, the precision recovering CD4+/FOXP3+/CD8a+, PD1+/PDL1+ and CD68+/PDL1+/CD8a+/PD1 + markers is 0.56, 0.61 and 0.48 respectively, even better than the precision recovering the individual markers. We further observe that the model predictions underscore the immuno-suppressive interaction between PD1+ and PDL1+. We see that tiles of PD1+:PDL1+ interactions are enriched for CD45+ ($3.0 * 10^{-16}$ one-sided paired t-test p-value) and depleted for CK + ($1.1 * 10^{-13}$ one-sided paired t-test p-value) in comparison to non-interacting tiles (S5 Fig). Moreover, the model highlights these trends even better than the tile level measurements for individual slices improving the statistical significance.

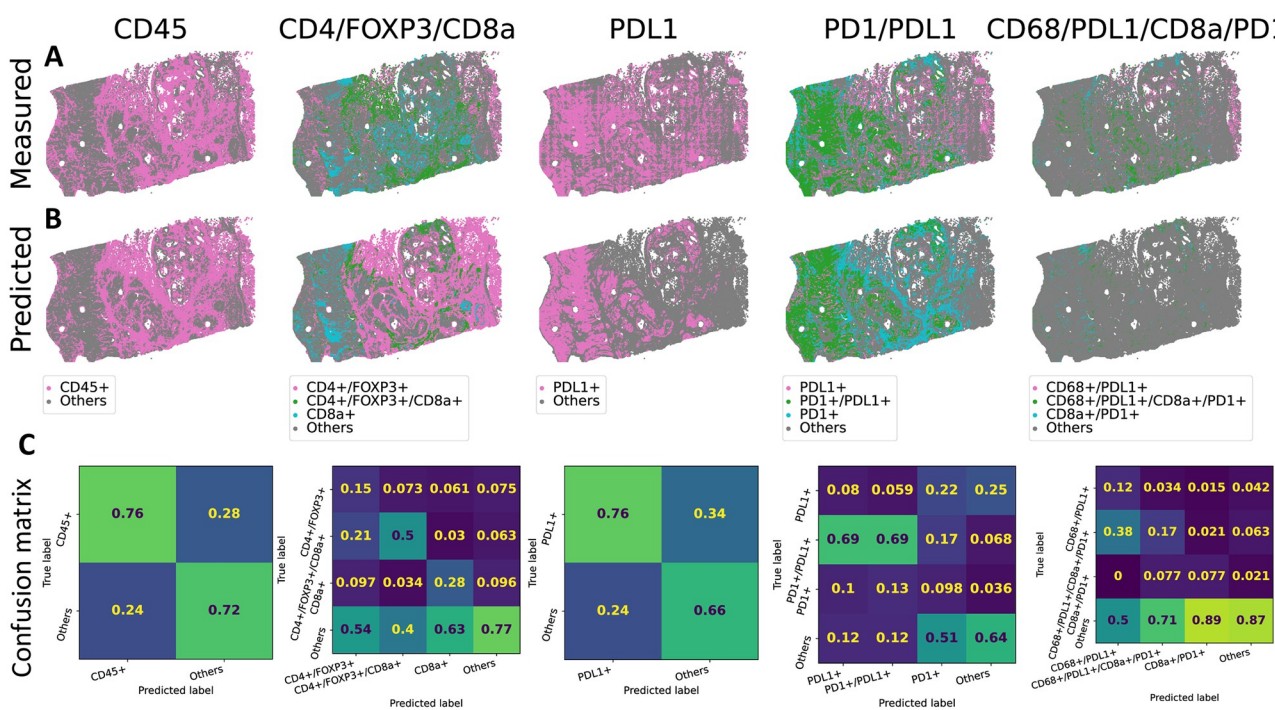

**Fig 5. Cell markers occurrence in CRC15.** See Fig 4 for panel description.

We further evaluated HIPI on marker prediction and co-occurrence on the external test CRC samples that were not used for training (Section 3.2). We see that the model is able to predict CD45 and PDL1 with 0.76 precision, removing unwanted artifacts in the measured data. Similarly, we are able to predict the PD1+/PDL1+ and CD4+/FOXP3+/CD8a+ interactions with 0.69 and 0.5 precision respectively (Fig 5). Since cell markers overlap, the model sometimes predicts a partial marker set. For example, out of the tiles HIPI predicted to be CD4+/FOXP3+, 21% were measured to be CD4+/FOXP3+/CD8a+. In addition, we see that HIPI captures the decrease of Keratin expression (0.03 one-sided paired t-test p-value) and the increase in CD45 expression (0.01 one-sided paired t-test p-value) in PD1+:PDL1+ interacting tiles in the new samples (S5 Fig). In contrast, the measured tile level calls do not show similar trends which further illustrates the advantage of out model.

## 3.4 Comparison to baseline models

In addition to HIPI, we also trained two *baseline* models to evaluate the use of a pathology specific self supervised pretrained feature extractor. The first baseline model, is a simple model that uses only the mean image intensities as the image features together with a linear layer as a prediction head. That is, the baseline model does not use the tissue morphology but only the color intensities in tiles. The second model uses a ResNet50 feature extractor pretrained on ImageNet1k [27] and the same MLP architecture as HIPI for prediction (S1 Appendix). We used the same training tiles from samples of CRC01 and evaluated the performance on the left out tiles and the tiles of the other CRC samples (Section 2.1).

We observe that our model which uses a pretrained SSL feature extractor, consistently gives higher correlation to the measured tile protein expression than both baseline model that uses only color intensities and the baseline model that uses a ResNet50 feature extractor (S1 Fig).

Each model shows similar correlation metrics between tiles from the training, validation and test sets of CRC01. Some markers are relatively easy to identify with relatively high correlation even with the simple average color model (S1 Fig). For example, the average color baseline model gives a median correlation of 0.7 on Keratin levels whereas the ResNet50 model gives a median correlation of 0.74 and the HIPI model gives a median correlation of 0.86. On the other hand, there are proteins that are relatively difficult to predict based only on an H&E image tile. For instance, prediction of CD31 expression has median correlations of 0.2 and 0.32 using the baseline models and a 0.45 median correlation using the HIPI model. For some proteins the HIPI model shows a significant advantage. For example, the baseline models gives median correlations of of 0.18 and 0.47 predicting CD45 while the HIPI model gives a 0.65 median correlation (S1 Fig). Surprisingly, on some markers, such as Ki67 and PDL1, the simple model based on average color outperforms the more complex model that uses a ResNet.

To evaluate the ability to generalize to new samples, we evaluate the baseline models performance on the tiles of the additional CRC samples [25]. We see that our HIPI model is able to generalize to new samples correctly predicting tile level expression of several proteins (S2 and S3 Figs). HIPI gives higher correlations to measured values on all markers and samples than both baseline models. While some proteins can be predicted on the new samples at similar levels of correlation as the training samples (e.g. Keratin, Desmin Ki67, CD45), other proteins are more difficult to predict (e.g. CD3, CD20, CD68). Still, we see that our model is able to better predict tile protein expression than the baseline models. This shows that the image tissue structure indeed provides valuable information and that using a pathology-designated feature extractor is important for obtaining meaningful representations for prediction of marker expression (S2 and S3 Figs).

## Discussion

We describe HIPI, a method for predicting protein markers measured with CyCIF from H&E images. We specifically described our results on colorectal cancer samples. We design a pipeline for aligning and processing paired CyCIF and H&E images of adjacent tumor slices. We then train a model, based on a dedicated pre-trained H&E image encoder, for predicting CyCIF protein expression from small image tiles. We evaluate our model on unseen image regions from samples used for training as well as new samples never used in training. This is in contrast to previous methods for generating spatial molecular measurements from H&Es, which only evaluate on samples or regions coming from the same samples or regions used for training their models; as a result, much less can be said about such methods' generalization abilities. On most measured proteins, our model is able to recover the expression levels on both internal and external evaluation samples. This shows that tissue morphology captured in the H&E image provides important information pertaining to the types of cells observed, the markers they express, and the extent of this expression. AI inference of protein expression from H&E can be used as a first stage assessment and that actual molecular assay of IHC staining can follow to obtain high confidence for the suggested findings.

Unlike CyCIF or other technologies, HIPI predicts marker expression at low resolution image tile level. This lower resolution tile level measurement is an aggregate of all cells in the frame. As a result, the expression of different cells may cancel one another and may hinder the ability to observe rare cell types. In addition, with tile level view we lose information that might be important in order to understand cell-cell interactions. On the other hand, using tile level measurements allows our method to cope with small misalignments between the adjacent tissue slices used for training. Moreover, tile level measurements reduce the variation observed in cell level data or the exact calls of cell positions.

Our method can be improved and generalized in several ways. First, while we used CyCIF, there are other spatial technologies that yield molecular measurements at different resolutions [28]. Although methods have been proposed to generate spatial transcriptomics measurements from H&E images, they do not generalize to out of distribution samples [17]. Second, although we used our model to train and infer on colorectal cancer samples, we could train and generalize to other cancer types. In fact, the cancer type, as well as other clinical information can be incorporated into the model to discern inter-sample variability. Third, it might be useful to improve our spatial resolution to allow single cell resolution instead of tile level resolution. This requires better cell segmentation from both the H&E and the CyCIF images and calling the cell level CyCIF protein expressions. Moreover, it requires aligning not only the images but also the cells between two (slightly) different tissue slices.

## Supporting information

**S1 Appendix. Supplementary methods and analyses.** More details on the method and implementation, together with additional figures.
(PDF)

**S1 Fig. Correlation between measured and predicted tile level expression for 22 samples from patient CRC01 [25].**
(TIF)

**S2 Fig. Correlation between measured and predicted tile level expression for patients CRC 2, 3, 12, 13, 14, 15 and 17 with different models. [25].**
(TIF)

**S3 Fig. Correlation between measured and predicted tile level expression for test tiles of CRC01 and patients CRC 2, 3, 12, 13, 14, 15 and 17 with different models. [25].**
(TIF)

**S4 Fig. HIPI top X% accuracy.**
(TIF)

**S5 Fig. The proportion of Keratin+ and CD45+ tiles out of PDL1+:PD1+ interacting and non-interacting tiles.**
(TIF)

**S1 Table. Top-20% accuracy.**
(PDF)

## Author Contributions

**Conceptualization:** Ron Zeira, Leon Anavy, Zohar Yakhini, Daniel Freedman.

**Data curation:** Ron Zeira.

**Formal analysis:** Ron Zeira.

**Funding acquisition:** Ehud Rivlin, Daniel Freedman.

**Investigation:** Ron Zeira.

**Methodology:** Ron Zeira, Leon Anavy, Zohar Yakhini, Daniel Freedman.

**Project administration:** Ehud Rivlin, Daniel Freedman.

**Resources:** Daniel Freedman.

**Software:** Ron Zeira.

**Supervision:** Ehud Rivlin, Daniel Freedman.

**Validation:** Ron Zeira.

**Visualization:** Ron Zeira.

**Writing – original draft:** Ron Zeira, Leon Anavy, Zohar Yakhini, Daniel Freedman.

**Writing – review & editing:** Ron Zeira, Leon Anavy, Zohar Yakhini, Daniel Freedman.

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
