## [Decision Letter · Decision Letter 0]

23 Aug 2024

Dear Dr. Zeira,

Thank you very much for submitting your manuscript "HIPI: Spatially Resolved Multiplexed Protein Expression Inferred from H&E WSIs" for consideration at PLOS Computational Biology. As with all papers reviewed by the journal, your manuscript was reviewed by members of the editorial board and by several independent reviewers. The reviewers appreciated the attention to an important topic. Based on the reviews, we are likely to accept this manuscript for publication, providing that you modify the manuscript according to the review recommendations.

In addition to the reviews you will see below, we have asked reviewers 1 and 2 to comment on your reply to reviewer 3, who was not available to re-review the manuscript. In your revision, please address these additional points as well:

Reviewer 1: 

I believe the authors have satisfactorily addressed Reviewer 3's comments.

The first comment is that the authors do not compare against existing methods, but the authors correctly note that the methods are different, and a direct comparison to these methods is not informative. They do add a comparison with other methods, partially in response to other reviewer feedback. More comparisons with existing methods would be welcome, but are not informative if they address different tasks.

 The second comment is that the authors do not describe the method adequately. It is described appropriately. The third comment is that the authors do not share their code. This was a significant problem that the other reviewers noted, but they have now shared their code.   Reviewer 2:  I had a quick look on the response to reviewers 3 and here are the comments for the 3 points, accordingly:

1) the benchmarking methods are indeed a good question. The authors now added more baseline methods, which are good, but saying existing RNA-prediction methods are readily applicable. This is true but not undoable at all. 

2) the methodology is indeed mainly within the supp S1. While concise, it looks major information is included. The authors may need to structure it to align with the journal format and for easier reading.

3) it looks the codes for the models are available but the analysis notebooks are not. It will be very helpful to ensure reproducibility if the authors can also release the analysis notebooks/scripts.

Sincerely,

Kamila Naxerova

Guest Editor

PLOS Computational Biology

Jason Papin

Editor-in-Chief

PLOS Computational Biology

Reviewer's Responses to Questions

**Comments to the Authors:**

Reviewer #1: The authors have satisfactorily addressed the comments from my previous review, with two minor exceptions.

1. The authors have provided their code, which is important and appreciated. I would encourage them to update the README in their GitHub repository to provide example commands for their entire algorithmic pipeline. Example commands for the training and inference steps are provided, but not for the data processing steps.

2. The authors have updated their figures to make them more legible, but there are still parts with small text that are difficult to read, namely, the color bars in Figures 2 and 3 and the legends in Figures 4 and 5. Unfortunately, the figures are not vector graphics, but, fortunately, they appear to be high-resolution raster graphics that can be enlarged on a computer screen and more easily read. I would encourage them to enlarge the small text in the figures so that it can be read by most people on a printed sheet.

Reviewer #2: I am the reviewer #2 from RECOMB. I thank the authors for replying to my comments and adding more analyses. Overall, I think this would be valuable work but some changes in the presentation may help avoid over-presenting the performance, therefore I would further clarify my original comments 3 & 4.

# 3. While using the consecutive section seemingly works well, it still has risks of over-connecting the two modalities during the registration. Therefore, if possible, please consider using at least one slide from the technology supporting within the slide (proteins and H&E) as a test set.

# 4. The new supp figures S2 and S3 strongly suggest that there is a substantial decrease when applying to external datasets. I’m not saying the performance is too low in the external datasets; it is still within a reasonable range, considering the limited accuracy in predicting RNA levels in spatial transcriptomics data. However, the much higher performance in the within-slide test set can be over-confident about the prediction, as the framework was proposed to predict the unseen H&E tissues, instead of unseen regions of the tissue. Please re-arrange the results by presenting the cross-slide or cross-sample prediction as the main figure.

**Have the authors made all data and (if applicable) computational code underlying the findings in their manuscript fully available?**

Reviewer #1: Yes

Reviewer #2: None

PLOS authors have the option to publish the peer review history of their article (what does this mean?). If published, this will include your full peer review and any attached files.

Reviewer #1: No

Reviewer #2: No

Figure Files:

Data Requirements:

Reproducibility:

References:

---

## [Editor Report · Decision Letter 1]

19 Sep 2024

Dear Dr. Zeira,

We are pleased to inform you that your manuscript 'HIPI: Spatially Resolved Multiplexed Protein Expression Inferred from H&E WSIs' has been provisionally accepted for publication in PLOS Computational Biology.

Best regards,

Kamila Naxerova

Guest Editor

PLOS Computational Biology

Jason Papin

Editor-in-Chief

PLOS Computational Biology

---

## [Editor Report · Acceptance letter]

23 Sep 2024

PCOMPBIOL-D-24-01094R1 

HIPI: Spatially Resolved Multiplexed Protein Expression Inferred from H&E WSIs

Dear Dr Zeira,

I am pleased to inform you that your manuscript has been formally accepted for publication in PLOS Computational Biology. Your manuscript is now with our production department and you will be notified of the publication date in due course.

With kind regards,

Zsofia Freund
